# Assessment of Perceived Health Status and Access to Health Service during the COVID-19 Pandemic: Cross-Sectional Survey in Italy

**DOI:** 10.3390/vaccines10122051

**Published:** 2022-11-30

**Authors:** Concetta Paola Pelullo, Pamela Tortoriello, Silvia Angelillo, Francesca Licata, Francesco Napolitano, Gabriella Di Giuseppe

**Affiliations:** 1Department of Movement Sciences and Wellbeing, University of Naples “Parthenope”, 80133 Naples, Italy; 2Department of Experimental Medicine, University of Campania “Luigi Vanvitelli”, 80138 Naples, Italy; 3Department of Health Sciences, University of Catanzaro “Magna Græcia”, 88100 Catanzaro, Italy

**Keywords:** COVID-19, cross-sectional, perceived health status, health services, SF-12

## Abstract

(1) Background: The aims of this survey were to assess the perceived health status and to evaluate the use of healthcare services during the pandemic period. (2) Methods: This cross-sectional survey was conducted from May to October 2021 in the Campania and Calabria regions, Southern Italy. The sample was selected among 655 subjects attending vaccination or primary care physician clinics. (3) Results: More than half (57.2%) of the respondents were female, the mean age was 49 years (range 19–97), and 61.3% had at least one chronic disease. Among the respondents, 56.3% declared that they had accessed healthcare at least once during the pandemic and, among all respondents who did not access healthcare, 23.2% gave reasons related to the COVID-19 pandemic. The two scores obtained from the Short-Form-12 Health Survey (SF-12), physical health summary (PCS) and mental health summary (MCS), had a means of 48.4 and 45.9, respectively. Among the respondents, 2.3% of respondents considered their health poor, 43.1% good and 6.4% excellent. (4) Conclusions: Our results suggest the need to ensure, in similar health emergency situations, a quick response from the National Health System so that ordinary medical assistance activities can be guaranteed in full safety, avoiding the risk of missed access or lack of assistance.

## 1. Introduction

As is already known, the Coronavirus Disease 2019 (COVID-19) epidemic began in Wuhan, China, in December 2019, and then spread across several countries and affected a large number of people, before eventually being classified as a pandemic. In Italy, on 9 March 2020, the Ministry of Health, to contain the spread of contagion, ordered a national lockdown. As has already been shown in other studies conducted during the health emergency, the fear of contracting an unknown infection can affect the way in which health services are used and reduce access to healthcare for the population in need of treatment [1,2,3]. The provision of non-urgent classified care has therefore been hampered by the pandemic crisis and isolation [4,5,6,7,8]. However, it is not known whether access to health and care services during the COVID-19 pandemic differed according to previously known factors, such as gender, age, education level, social relationships, presence of medical problems, race or ethnic group, socio-economic and employment status [9,10,11,12,13], or expanded further existing health inequalities. Evidence from previous pandemics suggests this possibility, but data in the context of the COVID-19 pandemic are limited [14,15,16]. Furthermore, little is known about the general health of the population during difficult times such as the COVID-19 pandemic [17,18]. Although several investigations in the literature have evaluated access to health services [19,20,21] and the perceived health status in different populations such as adults [22], parents [23,24], the incarcerated [25,26,27], very little information is available regarding the access to health services and perceived health status during the COVID-19 lockdown [28,29,30,31,32].

Therefore, the aims of this survey were: (1) to assess the perceived health status; (2) to evaluate use of healthcare services; (3) and to investigate any barriers to the general population accessing health services during the pandemic period.

## 2. Materials and Methods

### 2.1. Study Population and Sample Size

This cross-sectional survey was conducted from May to October 2021 in the Campania and Calabria Regions (Southern Italy). The sample was selected among subjects attending vaccination or primary care physician clinics.

Participants needed to be at least 18 years of age to be eligible. The sample size was calculated before the beginning of the study, assuming that 50% of respondents had a high perception level of health status, a confidence interval of 95%, a margin of error of 5%, and a response rate of 70%. Therefore, the sample size was estimated to be 549 participants.

### 2.2. Instrument

The questionnaire was constructed considering an extensive literature review and the content was developed from several validated sources used to investigate access to healthcare services for non-communicable diseases by children and adolescents [21], access to health services among prisoners [20,27] and other populations [19]. In particular, the questionnaire was composed of the five following sections: (1) perceived health status; (2) anamnestic characteristics (history of chronic diseases such as diabetes, hypertension, heart attack, angina pectoris, osteoarthritis/arthritis, lumbosciatalgia, cancer, migraine, chronic bronchitis, peptic ulcer, prostatic hypertrophy, and nervous disorders including Parkinson’s, Alzheimer’s, epilepsy, etc.), prescribed pharmacological therapies and their duration; (3) access to health services during the pandemic (containing questions about the clinical characteristics of the chronic diseases, health problems during the pandemic, perceived severity of health problems and related medical visits, primary care physician accesses, specialists visits, emergency accesses, hospital admissions, and integrated home care, reasons for not having used healthcare such as no health problems, minor health problems, fear of contracting COVID-19, etc.); (4) source of information about COVID-19 and need for additional information; and (5) socio-demographic characteristics (age, gender, marital status, number of children, number of cohabitants, education level, working activity). In particular, to assess perceived health status, the Short-Form-12 Health Survey (SF-12) was used, which consists of 12 items derived from an extended version called SF-36 (consisting of 36 items). It provides two parameters measuring both physical (physical health summary—PCS) and mental health (mental health summary—MCS), allowing to assess the overall self-perceived health status [33,34,35]. This survey has been widely applied around the world in several studies involving sick or healthy population samples, showing remarkable effectiveness in all groups thanks to its understandability and shortness [36,37,38,39,40].

The study protocol and the questionnaire were approved by the Ethics Committee of the Teaching Hospital of the University of Campania “Luigi Vanvitelli” (n 021807/i 2021).

Before the beginning of the study, the questionnaire was pilot tested on 50 subjects in order to ensure the readability and reliability of the questions.

### 2.3. Data Collection 

Data collection was conducted by physicians not involved in patient care. The researchers previously informed all selected subjects about the survey objectives, specifying that participation was voluntary and confidential, and ensuring that personal information was not contained in the questionnaire. From each participant, verbal informed consent was requested; those who agreed to participate then completed a self-administered questionnaire. Subjects who were unable to read, understand the Italian language, or give informed consent due to cognitive impairment were excluded from the survey.

### 2.4. Statistical Analysis

The data were analyzed with descriptive (frequencies, means, standard deviations) and inferential (bivariate and multivariate analysis) statistics using the Stata software version 15 [41]. Bivariate analysis, using the chi-squared test, Student’s *t*-test, and the analysis of variance (ANOVA), was performed in order to evaluate the relevant differences in the mean of PCS-12 and MCS-12 due to several characteristics. The independent variables with a significance level of *p*-value less than or equal to 0.25 were then included in multivariate linear and logistic regression models with the purpose of identifying those predicting the outcomes of interest: physical health summary—PCS (continuous) (Model 1); mental health summary—MCS (continuous) (Model 2); and medical examinations and/or tests during the pandemic (no = 0, yes = 1) (Model 3).

The following explanatory variables were included in all multivariate linear and logistic regression models: gender (male = 0, female = 1); age in years (continuous); marital status (unmarried/other = 0, married = 1); education level (no formal education/elementary school/middle school = 1, high school = 2, university degree = 3); working activity (no = 0, yes = 1); hypertension (no = 0, yes = 1); diabetes (no = 0, yes = 1); heart diseases (no = 0, yes = 1); neuropsychiatric diseases (no = 0, yes = 1); musculoskeletal diseases (no = 0, yes = 1); other diseases (benign prostatic hyperplasia, endocrine diseases, immunological diseases) (no = 0, yes = 1); and the need for additional information about COVID-19 (no = 0, yes = 1). Moreover, in Models 1 and 2, the variables medical examinations and/or tests during the pandemic (no = 0, yes = 1), number of cohabitants (continuous), accesses to a primary care physician (PCP) (none = 0, ≥1 = 1), specialists visits (none = 0, ≥1 = 1), emergency accesses (none = 0, ≥1 = 1), and hospital admissions (none = 0, ≥1 = 1) were also included. In Model 3, the variables physical health summary—PCS (continuous), mental health summary—MCS (continuous), number of chronic diseases (<3 = 0, ≥3 = 1), number of drugs (<3 = 0, ≥3 = 1), clinical characteristics of chronic diseases during the pandemic (improved/stable = 0, worsened = 1), how long one had been suffering from a chronic disease (<9 years = 0, ≥9 years = 1), health problems during the pandemic (no = 0, yes = 1), perception of the health problem (very slight/slight = 0, moderate/severe/very severe = 1), physician as a source of information about COVID-19 (no = 0, yes = 1), number of children (none = 0, ≥1 = 1) and number of cohabitants (none = 0, ≥1 = 1) were also included.

Backward stepwise procedures were applied, including in the final models where the only characteristics provided a significant explanation of the outcomes, with a threshold of *p*-values of 0.2 for entering and of 0.4 for being retained. Adjusted odds ratios (ORs) and 95% Confidence Intervals (CIs) were presented in the logistic regression models and standardized regression coefficients (β) in the linear regression models. All analyses were two-sided and the level of statistical significance was set at *p* equal or to or less than 0.05.

## 3. Results

Of the 680 subjects that were approached, a total of 655 agreed to participate for a response rate of 96.2%. Table 1 summarizes the main characteristics of the sample. More than half (57.2%) of the respondents were female, the mean age was 49 years (range 19–97), 55.6% were married, 64.1% had more than one cohabitant, 52.7% had a high school degree, and a little more than half (51.9%) were unemployed. Moreover, 38.7% of participants did not have any chronic disease, 61.3% had at least one and, among them, 26.1% had two and 20.3% had three or more chronic diseases. In particular, 24.8% had hypertension, 14% diabetes, 12.7% musculoskeletal diseases, 7.9% heart diseases, 5.4% neuropsychiatric diseases, and 14.4% other diseases. 

Among the participants who had chronic diseases, 12.1% reported the worsening of clinical conditions during the pandemic. Furthermore, 23.8% of all respondents had a health problem during the pandemic, and 38.3% of them perceived it as a moderate problem, 32.2% as a serious problem, and 8.1% as a very serious one. Among those who had a health problem during the pandemic, 48.6% initially contacted their PCPs, 20.3% contacted specialists, and 20.9% went to the emergency room.

The results of the PCS-12 and MCS-12 summary scores, and of the SF-12 items are presented in Table 1 and Table 2, respectively. The overall mean PCS-12 was 48.4 (SD ± 10.2; median = 51.4) and the overall mean MCS-12 was 45.9 (SD ± 10.4; median = 48.2).

Among the respondents, 2.3% considered their health “poor”, 43.1% “good” and 6.4% “excellent”; 3.8% of participants declared their health “limited them a lot” during typical daily activities, and 26.3% that their health “limited them a little”. During the previous four weeks, 22.1% “were limited in the kind of work or other activities” because of their health, 28.8% “did work or activities less carefully than usual as a result of emotional problems (such as feeling depressed or anxious)”, and 58.5% of all respondents affirmed that “pain interfered with their normal work (including work outside the home and housework)” from a “little bit” (21.4%) to “extremely” (1.4%). Only 7.2% of participants “felt calm and peaceful all of the time” during the previous four weeks and 0.8% “none of the time”; 5.5% “had a lot of energy all of the time” and 1.7% “none of the time”; 0.8% “felt downhearted and blue all of the time” and 8.2% “none of the time”. Moreover, during the previous four weeks, 1.8% of respondents declared that “their physical health or emotional problems interfered with their social activities all of the time” and 28.7% “none of the time” (Table 2).

Table 1 and Table 3 show the results of the univariate and multivariate analyses of outcomes of interest. Univariate analysis has evidenced significant differences according to several characteristics. In particular, PCS-12 was lower in males, in the elderly, in those who were married, in those with a lower education level, in people with chronic diseases such as hypertension, diabetes, heart diseases, musculoskeletal diseases, neuropsychiatric diseases, and among subjects who accessed health facilities (PCPs, specialists, emergency and hospital accesses). As regards MCS-12, it was significantly lower in females, younger respondents, in those who were unmarried, in those with fewer cohabitants, in those suffering from neuropsychiatric diseases, and in those who accessed to the emergency room. 

The results of the multivariate analysis confirm that the independent variables improving PCS-12 (Model 1 in Table 3) score were decreasing age, absence of musculoskeletal diseases, heart diseases, neuropsychiatric diseases, diabetes and hypertension, no access to healthcare, and higher education level. However, MCS-12 (Model 2 in Table 3) was improved in males, in those who were married, in the absence of neuropsychiatric diseases, in those who did not access the emergency room, in those who did not need further information on COVID-19, and in those with fewer cohabitants.

When asked about their access to healthcare during the pandemic period, 56.3% of respondents declared that they had at least one access, in particular, 40.2% to the PCP, 33.2% to specialists, 6.7% to emergency room, and 6.8% declared having been hospitalized. 

Of all respondents who declared that they did not access healthcare during the pandemic, with or without health problems, 23.2% gave reasons related to the COVID-19 pandemic. 

In Model 3 (Table 3), medical examinations and/or tests during the pandemic were carried out by individuals who suffered from health problems (OR = 22.73; 95% CI 8–64.55), people with a lower PCS-12 score (OR = 0.96; 95% CI 0.93–0.98), and in those who had children (OR = 1.85; 95% CI 1.04–3.28).

Overall, only 33% of respondents had received information about COVID-19 from their physicians, 77.1% from the Internet, 62.9% from television, and 16.9% from newspapers. Moreover, 40% declared that they felt a need for additional information about COVID-19.

## 4. Discussion

To the best of our knowledge, this survey is one of the few that offers an assessment of the perceived health status of the general population during the pandemic period, also evaluating the use of healthcare services.

The most remarkable results from our sample show an overall mean of 48.4, and of 45.9 for PCS-12 and MCS-12, respectively. These values, compared to the national ISTAT data on the general population reported in 2013, are lower by 2.3 and 3 points (mean PCS-12 = 50.7, mean MCS-12 = 48.8), respectively [42]. This also means that the gap between the physical and the mental component summary that emerged from our study is of 2.5 points, which is higher than the Italian general population norm (1.9 points) documented in ISTAT report. This difference could be attributed to the greater sensitivity of mental health, compared to physical health, to these kinds of crisis events such as a pandemic. As is known, Italy is one of the countries affected earlier and more intensely by the pandemic due to COVID-19. Indeed, the rapid evolution of the health emergency and its burden on social behavior and on the National Health System (NHS) had an immediate impact in terms of mental and physical health. On the one hand, the governments of the most affected countries implemented a series of actions to mitigate the spread of infections and to reduce the consequent pressure on the hospital system. On the other hand, the COVID-19 pandemic has caused a series of other cascading psychological effects which will probably be much more difficult to mitigate and which expose the younger and more vulnerable sections of the population to complex consequences; therefore, it is urgent to prepare and expand access to care, starting with the very young and frail.

Several studies have been carried out around the world during the lockdown period and restrictions that almost unanimously describe the heavy psychological impact that the pandemic has had on individuals.

Recently, precisely because of the alarms raised by numerous psychologists, a debate on the subject has ignited which aims above all to promote the implementation of programmatic action to widen access to treatment for individuals most at risk, typically the very young and workers who are in a state of precariousness and who, due to low salaries, have difficulty accessing psychological therapies by resorting to the private sector. These are the subjects who have suffered the most from the psychological impact of the pandemic, also developing disorders such as anxiety, panic attacks, and depression. The NHS must prepare for the psychological effects deriving from the pandemic, with problems such as anxiety and depression so far contained by the emergency context itself but ready to manifest their long-term effects. The comparison with other cross-sectional surveys, conducted during the pandemic, showed that the Chinese adult population had PCS-12 and MCS-12 that are quite higher than Italian ones (mean PCS-12 = 75.3, mean MCS-12 = 66.6), despite the fact that our data were collected during a different pandemic period characterized by fewer restrictions than the first lockdown (February—March 2020 vs. May—October 2021) [17]. Moreover, another study conducted among Turkish respondents showed higher physical and mental health scores (mean PCS-12 = 73, mean MCS-12 = 52.4) than Italians [43]. Another study assessed PCS-12 and MCS-12 in Italy during the pandemic among college students and, as expected, found a higher physical score (mean PCS-12 = 54.3), usual in younger people [35], and a lower mental score, similar to that of the younger age group of our sample (mean MCS-12 = 41.4 vs. mean MCS-12 = 41.6) [40]. By contrast, only one survey, carried out during the pandemic among citizens of Saudi Arabia aged 18 years and more, showed mean values of PCS-12 and MCS-12 lower than our Italian data [44]. Some of these differences in self-perceived health among countries could be attributed to different welfare regimes and population-reporting styles [45,46], and to the way in which the different countries have been affected and responded to the pandemic emergency situation. Despite the sometimes scarce economic resources allocated to it in the last decade, the NHS has been able to react with commitment and competence, even if sometimes in trouble and with difficulty, to the call of the emergency. Hospitals have been subjected to unprecedented pressure and the effect has reverberated as represented by the decrease in hospitalizations for cardiovascular and cerebrovascular disease, except that for non-deferrable oncological and orthopedic operations, and access to cancer prevention screening.

Furthermore, this research also analyzed all single SF-12 items: in particular, only 7.2% of participants “felt calm and peaceful all of the time” during the previous four weeks. This value is lower than that reported in a similar survey conducted in Italy in 2013 [22]. This effect on mental perceived health may reflect possible higher stress levels than in non-emergency periods.

In line with other studies, in our survey, the independent variables significantly improving PCS-12 were decreasing age, absence of musculoskeletal diseases, heart diseases, neuropsychiatric diseases, diabetes and hypertension, no access to healthcare, and higher education level, whereas MCS-12 was improved by being a man, the absence of neuropsychiatric diseases, and the absence of emergency accesses [22,35]. In our study, MCS-12 was also improved in those who were married, in those who did not need further information on COVID-19 and in those with fewer cohabitants. Even these variables might be influenced by the stress linked to the fear of contagion. Furthermore, some studies conducted in Italy have considered the effects of COVID-19 on the psychic sphere, with a deterioration in relationships with partners, with their children, and an increase in perceived fatigue during the performance of work activities [47,48].

Our findings show that 56.3% of participants had at least one access to healthcare during the pandemic period. Among the respondents who did not have access to healthcare during the pandemic, neither for health problems nor for routine checks, 23.2% reported reasons related to the COVID-19 pandemic. In line with another cross-sectional survey that investigated the same issues in Italy during the pandemic, such as research by Gualano et al., in which it was observed that 32.4% of respondents faced a delay of a scheduled medical service by provider decision and 13.2% refused to access scheduled medical service due to the fear of contagion [2]. Although our results are numerically less impressive, considering the different period during which the data were collected, the problem of delaying and postponing medical examinations lasted also during the second pandemic year, so it could represent a problem in any case of future emergency situations. 

However, unsurprisingly, medical examinations and/or tests during the pandemic were carried out more significantly by respondents suffering from health problems, people with a lower PCS-12 score, and those who had children. Definitely, as already mentioned, one of the factors increasing PCS-12 was the lack of access to health services, and, at the same time, medical examinations and/or tests during the pandemic were carried out by individuals who had a lower PCS-12 score. As documented in the 2013 ISTAT report, Italy has an aging population, in which chronic diseases are increasingly common. Compared to 2005, chronic respiratory diseases and arthrosis have decreased while malignant tumors, Alzheimer’s, and senile dementias have increased. Despite this, physical health improves, while mental health worsens compared to 2005: the latter decreases on average by 1.6 points, especially among young people up to 34 years of age (−2.7 points), especially males, and among adults aged 45–54 (−2.6). Therefore, as also shown in the previous literature, it can be affirmed that perceived mental and physical health status might reflect the need for health services among the population [22,49]. 

It is essential to read the findings from this research in light of some possible limitations. First of all, cross-sectional investigations have, as intrinsic problems, that of not allowing cause–effect associations because independent variables and outcomes are simultaneously evaluated. Secondly, all information was self-reported: this is mandatory for collecting data on perceived health status, but it could represent a problem in the field of access to healthcare due to the potential recall bias. Additionally, a social desirability bias might be possible, although the anonymity of the survey allows for the minimal probability of this.

## 5. Conclusions

In conclusion, our findings showed a low score of PCS-12 and MCS-12, which should be of interest to policymakers. Moreover, considering that some of these findings indicate that the effect of pandemic stress can also influence the behavior of the population in terms of health, our results suggest the need to ensure, in similar health emergency situations, a quick response from the NHS so that ordinary medical assistance activities can be guaranteed in full safety, avoiding the risk of missed accesses or lack of assistance. Furthermore, perceived mental and physical health status may reflect the need for health services among the population. Thus, in this context, it is very useful to collect data on the utilization of health services and self-perceived health in order to offer better medical and psychosocial assistance where needed, during similar emergency situations. Therefore, future research after the COVID-19 pandemic should be carried out in order to address differences before and after this period, in order to identify the possible determinants of physical and mental health. 

## Figures and Tables

**Table 1 vaccines-10-02051-t001:** SF-12 Physical Component Summary (PCS) and Mental Component Summary (MCS) according to socio-demographic and anamnestic characteristics of the study population.

Socio-Demographic and Anamnestic Characteristics	N (%)	PCS Mean (±SD)	MCS Mean (±SD)
**Gender**			
Man	280 (42.8)	47.2 (±10.29)	48.5 (±9.52)
Woman	374 (57.2)	49.3 (±10.04)	43.9 (±10.66)
		*t-test (652) = −2.62 p = 0.0089*	*t-test (652) = 5.66 p < 0.001*
**Age, years**	49 ± 19.8 (19–93) *		
18–40	235 (35.9)	54 (±6.13)	41.6 (±10.43)
41–60	195 (29.8)	48.5 (±8.88)	49.4 (±9.72)
>60	224 (34.3)	42.4 (±11.27)	47.2 (±9.46)
		*F-test (2, 651) = 96.04 p < 0.001*	*F-test (2, 651) = 36.64 p < 0.001*
**Marital status**			
Married	363 (55.6)	45.4 (±8.6)	48.4 (±9.41)
Others	290 (44.4)	52.1 (±10.42)	42.7 (±10.73)
		*t-test (651) = 8.8 p < 0.001*	*t-test (651)= −7.26 p < 0.001*
**Number of cohabitants**			
None	47 (7.2)	47.5 (±11.36)	46.4 (±11.14)
1	186(28.7)	43.6 (±11.45)	47.4 (±9.97)
>1	416 (64.1)	50.6 (±8.66)	45.1 (±10.41)
		*F-test (2, 646) = 32.66 p < 0.001*	*F-test (2, 646) = 3.16 p = 0.0431*
**Education level**			
No formal education/elementary school/middle school	106 (16.2)	41.6 (±12.13)	48.4 (±9.18)
High school	344 (52.7)	49.8 (±9.41)	44.1 (±10.62)
University degree	203 (31.1)	49.7 (±8.79)	47.5 (±10.2)
		*F-test (2, 650) = 31.25 p < 0.001*	*F-test (2, 650) = 10.91 p < 0.001*
**Working activity**			
No	331 (51.9)	48.3 (±11.19)	43.2 (±10.55)
Yes	307 (48.1)	48.4 (±9.08)	48.5 (±9.51)
		*t-test (636) = 0.2 p = 0.835*	*t-test (636) = −6.68 p < 0.001*
**Chronic diseases**			
No	249 (38.7)	54.2 (±5.4)	44.3 (±10.74)
Yes	395 (61.3)	44.7 (±10.86)	47.02 (±9.91)
		*t-test (642) = 12.81 p < 0.001*	*t-test (642) = −3.24 p = 0.001*
**Hypertension**			
No	484 (75.2)	50.4 (±9.04)	45.5 (±10.64)
Yes	160 (24.8)	42 (±11.04)	47.5 (±9.13)
		*t-test (642) = 9.63 p < 0.001*	*t-test (642) = −2.1 p = 0.035*
**Diabetes**			
No	554 (86)	49.3 (±9.89)	45.7 (±10.51)
Yes	90 (14)	42.5 (±10.46)	47.5 (±8.91)
		*t-test (642) = 6.02 p < 0.001*	*t-test (642) = −1.49 p = 0.136*
**Heart diseases**			
No	593 (92.1)	49.3 (±9.64)	45.7 (±10.44)
Yes	51 (7.9)	37.9 (±11.24)	48.9 (±8.22)
		*t-test (642) = 7.96 p < 0.001*	*t-test (642) = −2.14 p < 0.032*
**Musculoskeletal diseases**			
No	562 (87.3)	49.8 (±9.36)	45.9 (±10.47)
Yes	82 (12.7)	38.6 (±10.79)	46.61 (±9.22)
		*t-test (642) = 9.86 p < 0.001*	*t-test (642) = −0.58 p = 0.558*
**Neuropsychiatric diseases**			
No	609 (94.6)	48.7 (±10.1)	46.2 (±10.16)
Yes	35 (5.4)	43.1 (±11.38)	41.5 (±11.96)
		*t-test (642) = 3.14 p = 0.001*	*t-test (642) = 2.66 p = 0.007*
**Other diseases**			
No	551 (85.6)	48.85 (±10.1)	45.96 (±10.35)
Yes	93 (14.4)	45.52 (±10.66)	46.16 (±10.16)
		*t-test (642) = 2.91 p = 0.003*	*t-test (642) = −0.17 p = 0.864*
**Access to healthcare during the pandemic**			
No	282 (43.7)	51.5 (±9.08)	46.2 (±10.11)
Yes	364 (56.3)	45.9 (±11.01)	45.7 (±10.62)
		*t-test (644) = 7.22 p < 0.001*	*t-test (644) = 0.6 p = 0.544*
**Primary care physician (PCP) accesses**			
None	385 (59.8)	50 (±9.26)	45.8 (±10.25)
<5	191 (29.6)	48.3 (±9.69)	45.4 (±10.75)
≥5	68 (10.6)	38.8 (±11.76)	48 (±10.2)
		*F-test (2, 641) = 38.99 p < 0.001*	*F-test (2, 641) = 1.65 p = 0.1925*
**Specialists visits**			
None	430 (66.8)	49.6 (±9.67)	46.4 (±10.27)
≥1	214 (33.2)	45.8 (±10.83)	45 (±10.58)
		*t-test (642) = 4.5 p < 0.001*	*t-test (642) = 1.62 p = 0.105*
**Emergency accesses**			
None	600 (93.3)	48.8 (±10.04)	46.2 (±10.22)
≥1	43 (6.7)	41.5 (±10.56)	42.33 (±11.94)
		*t-test (641) = 4.64 p < 0.001*	*t-test (641) = 2.36 p = 0.018*
**Hospital admissions**			
None	601 (93.2)	48.5 (±10.18)	46 (±10.41)
≥1	44 (6.8)	45.7 (±10.6)	44.33 (±10.32)
		*t-test (643) = 1.79 p = 0.073*	*t-test (643) = 1.05 p = 0.29*

* Mean ± standard deviation (range).

**Table 2 vaccines-10-02051-t002:** SF-12 health survey.

SF-12 items	N (%)					
**This survey asks for your views about your health. This information will help keep track of how you feel and how well you are able to do your usual activities. Answer each question by choosing just one answer. If you are unsure how to answer a question, please give the best answer you can.**	**Excellent**	**Very good**	**Good**	**Fair**	**Poor**	
In general would you say your health is:	42 (6.4)	194 (29.6)	282 (43.1)	122 (18.6)	15 (2.3)	
**The following questions are about activities you might do during a typical day. Does your health now limit you in these activities? If so, how much?**	**YES, limited a lot**	**YES, limited a little**	**NO, not limited at all**			
2. Moderate activities such as moving a table, a vacuum cleaner, bowling, or playing golf.	25 (3.8)	172 (26.3)	458 (69.9)			
3. Climbing several flights of stairs.	27 (4.1)	162 (24.7)	466 (71.2)			
**During the past 4 weeks, have you had any of the following problems with your work or other regular daily activities as a result of your physical health?**	**YES**	**NO**				
4. Accomplished less than you would like.	144 (22)	511 (78)				
5. Were limited in the kind of work or other activities.	145 (22.1)	510 (77.9)				
**During the past 4 weeks, have you had any of the following problems with your work or other regular daily activities as a result of any emotional problems (such as feeling depressed or anxious)?**	**YES**	**NO**				
6. Accomplished less than you would like.	166 (25.3)	489 (74.7)				
7. Did work or activities less carefully than usual.	189 (28.8)	466 (71.2)				
8. **During the past 4 weeks, how much did pain interfere with your normal work (including work outside the home and housework)?**	**Not at all**	**A little bit**	**Moderately**	**Quite a bit**	**Extremely**	
	272 (41.5)	140 (21.4)	156 (23.8)	78 (11.9)	9 (1.4)	
**These questions are about how you have been feeling during the past 4 weeks. For each question, please give the one answer that comes closest to the way you have been feeling. How much of the time during the past 4 weeks**	**All of the time**	**Most of the time**	**A good bit of the time**	**Some of the time**	**A little of the time**	**None of the time**
9. Have you felt calm and peaceful?	47 (7.2)	187 (28.5)	141 (21.5)	235 (35.9)	40 (6.1)	5 (0.8)
10. Did you have a lot of energy?	36 (5.5)	129 (19.7)	162 (24.7)	239 (36.5)	78 (11.9)	11 (1.7)
11. Have you felt down-hearted and blue?	5 (0.8)	28 (4.3)	78 (11.9)	237 (36.2)	253 (38.6)	54 (8.2)
12. **During the past 4 weeks, how much of the time has your physical health or emotional problems interfered with your social activities (like visiting friends, relatives, etc.)?**	**All of the time**	**Most of the time**	**Some of the time**	**A little of the time**	**None of the time**	
	12 (1.8)	37 (5.6)	187 (28.6)	231 (35.3)	188 (28.7)	

**Table 3 vaccines-10-02051-t003:** Linear and logistic regression models.

**Model 1:** **Physical Health Summary—PCS** ***F* (10, 615) = 44.05; *R*^2^ = 0.42%; adjusted *R*^2^ = 0.41%;** ***p* < 0.0001**	**Coeff**	**SE**	** *t* **	** *p* **
*Age (* *continuous* *)*	−0.17	0.02	−6.54	<0.001
*Musculoskeletal diseases*				
No	1 ª			
Yes	−6.42	0.99	−6.44	<0.001
*Heart diseases*				
No	1 ª			
Yes	−5.98	1.23	−4.84	<0.001
*Access to healthcare during the pandemic*				
No	1 ª			
Yes	−2.26	0.67	−3.35	0.001
*Emergency accesses*				
No	1 ª			
Yes	−4.32	1.29	−3.35	0.001
*Diabetes*				
No	1 ª			
Yes	−2.72	0.93	−2.90	0.004
*Education level (ordinal)**	1.23	0.48	2.54	0.011
*Hypertension*				
No	1 ª			
Yes	−2.03	0.84	−2.42	0.016
*Neuropsychiatric diseases*				
No	1 ª			
Yes	−3.10	1.43	−2.16	0.031
*Marital status*				
Unmarried/other	1 ª			
Married	1.09	0.92	1.19	0.235
**Model 2: Mental health summary—MCS** *F* (9, 587) = 12.51; *R*^2^ = 0.16%; adjusted *R*^2^ = 0.15%; *p* < 0.0001	**Coeff**	**SE**	** *t* **	** *p* **
*Gender*				
Man	1 ª			
Woman	−3.71	0.8	−4.59	<0.001
*Marital status*				
Unmarried/other	1 ª			
Married	4.01	1.17	3.43	0.001
*Neuropsychiatric diseases*				
No	1 ª			
Yes	−6.00	1.76	−3.40	0.001
*Emergency accesses*				
None	1 ª			
≥1	−4.24	1.54	−2.75	0.006
*Need for additional information about COVID-19*				
No	1 ª			
Yes	−1.95	0.8	−2.44	0.015
*Number of cohabitants (continuous)*	−0.74	0.36	−2.01	0.045
*Education level (ordinal) **	0.64	0.59	1.08	0.278
*Age (* *continuous* *)*	0.03	0.03	0.97	0.334
*Heart diseases*				
No	1 ª			
Yes	1.45	1.5	0.96	0.335
**Model 3: Medical examinations and/or tests during the pandemic** Log likelihood = −179.05; χ^2^ = 104.01 (6 *df*); *p* < 0.0001	**OR**	**SE**	**95% CI**	** *p* **
*Health problem during the pandemic*				
No	1 ª			
Yes	22.73	12.1	8–64.55	<0.001
*Physical health summary* *(continuous)*	0.96	0.01	0.93–0.98	0.003
*Number of children*				
None	1 ª			
≥1	1.85	0.54	1.04–3.28	0.035
*Clinical picture of chronic diseases during the pandemic*				
Improved/stable	1 ª			
Worsened	1.98	1.01	0.72–5.38	0.181
*Number of drugs*				
<3	1 ª			
≥3	0.64	0.22	0.32–1.28	0.211
*Number of cohabitants*				
None	1 ª			
≥1	0.60	0.29	0.22–1.59	0.307

* Education level: (no formal education/elementary school/middle school = 1, high school = 2, university degree = 3). ª Reference category.

## Data Availability

The data presented in this study are available upon request from the corresponding author.

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
