# Peer review of "Assessment of Perceived Health Status and Access to Health Service during the COVID-19 Pandemic: Cross-Sectional Survey in Italy"

_vaccines, 2022, doi:10.3390/vaccines10122051_

Round 1

Reviewer 1 Report

The authors report on the perceived health of Italians during the 2nd wave of the COVID-19 pandemic, and work to identify which factors are most pronounced. Comparisons to similar reports from other countries, as well as the 2013 Italian study, provide some interesting comparisons. Overall the authors have done a solid job of presenting and discussing their data, and the final paragraph of the discussion addresses major shortcomings. Although not required, a more detailed comparison of this "snapshot" compared to the 2013 snapshot may be helpful in strengthening their argument regarding the national health service response during a pandemic if this comparison showed further variance from earlier studies (as compared to reports from other countries and their "better" scores).

Only a few minor edits needed such as lines 242 (in line whit) and line 271 (as intrinsic kind of problems). A little more reference to the 2013 study might provide more insight into deviations from baseline (or previous) values for PCS-12 and MCS-12, but the authors do discuss shortcomings of these types of studies in the final paragraph of the discussion.

Author Response

  1. The authors report on the perceived health of Italians during the 2nd wave of the COVID-19 pandemic, and work to identify which factors are most pronounced. Comparisons to similar reports from other countries, as well as the 2013 Italian study, provide some interesting comparisons. Overall the authors have done a solid job of presenting and discussing their data, and the final paragraph of the discussion addresses major shortcomings. Although not required, a more detailed comparison of this "snapshot" compared to the 2013 snapshot may be helpful in strengthening their argument regarding the national health service response during a pandemic if this comparison showed further variance from earlier studies (as compared to reports from other countries and their "better" scores). A little more reference to the 2013 study might provide more insight into deviations from baseline (or previous) values for PCS-12 and MCS-12, but the authors do discuss shortcomings of these types of studies in the final paragraph of the discussion.

As suggested, we have added in Discussion section a more detailed comparison with the study conducted in 2013 in Italy.

  1. Only a few minor edits needed such as lines 242 (in line whit) and line 271 (as intrinsic kind of problems).

As suggested, we have made the required changes.

Reviewer 2 Report

my opinion is that this paper should be sent to a journal for public health or another MDPI group because for a vaccine journal this paper has nothing to do with vaccines

Author Response

My opinion is that this paper should be sent to a journal for public health or another MDPI group because for a vaccine journal this paper has nothing to do with vaccines.

After the editor’s invitation, we proposed the title and the abstract to him and he expressed a positive opinion on the topic.

Reviewer 3 Report

COMMENTS FROM REVIEWER:

First and foremost, please accept my gratitude for the chance to evaluate this article.

The article "Assessment of perceived health status and access to health service during the COVID-19 pandemic: cross-sectional survey in Italy" delves into how to assess perceived health status and to evaluate the use of health care services during the pandemic period.

Overall, I have no reservations about the English language, as well as the methods and edited manuscript materials that have been rigorously edited. However, the parts on discussion and conclusion should be heavily altered in light of the research findings. The discussion part should be included in the final product, and please provide more paragraphs of discussion points. The importance of the survey and research projection, as well as the findings, must be included in the conclusion section.

The tables display many font styles. Before final publication, please double-check the format and typefaces according to the journal author handbook.

Author Response

First and foremost, please accept my gratitude for the chance to evaluate this article.

The article "Assessment of perceived health status and access to health service during the COVID-19 pandemic: cross-sectional survey in Italy" delves into how to assess perceived health status and to evaluate the use of health care services during the pandemic period.

Overall, I have no reservations about the English language, as well as the methods and edited manuscript materials that have been rigorously edited. However, the parts on discussion and conclusion should be heavily altered in light of the research findings. The discussion part should be included in the final product, and please provide more paragraphs of discussion points.

            As suggested, we have added more paragraphs in Discussion section.

The importance of the survey and research projection, as well as the findings, must be included in the conclusion section.

  As suggested, we have made the required changes.

The tables display many font styles. Before final publication, please double-check the format and typefaces according to the journal author handbook.

  As suggested, we have made the required changes.

Round 2

Reviewer 3 Report

The revision work has been completed in accordance with the reviewers' suggestions.

I would suggest that this work be considered for acceptance.